# A Black-Box Attack on Code Models via Representation Nearest Neighbor Search

Jie Zhang[1*], Wei Ma[2†], Qiang Hu[3], Shangqing Liu[2], Xiaofei Xie[4], Yves Le Traon[3], and Yang Liu[2]

[1]Noah's Ark Lab, Huawei
[2]School of Computer Science and Engineering, Nanyang Technological University
[3]The Interdisciplinary Centre for Security, Reliability and Trust, University of Luxembourg
[4]School of Computing and Information Systems, Singapore Management University

## Abstract

Existing methods for generating adversarial code examples face several challenges: limited availability of substitute variables, high verification costs for these substitutes, and the creation of adversarial samples with noticeable perturbations. To address these concerns, our proposed approach, RNNS, uses a search seed based on historical attacks to find potential adversarial substitutes. Rather than directly using the discrete substitutes, they are mapped to a continuous vector space using a pre-trained variable name encoder. Based on the vector representation, RNNS predicts and selects better substitutes for attacks. We evaluated the performance of RNNS across six coding tasks encompassing three programming languages: Java, Python, and C. We employed three pre-trained code models (Code-BERT, GraphCodeBERT, and CodeT5) that resulted in a cumulative of 18 victim models. The results demonstrate that RNNS outperforms baselines in terms of ASR and QT. Furthermore, the perturbation of adversarial examples introduced by RNNS is smaller compared to the baselines in terms of the number of replaced variables and the change in variable length. Lastly, our experiments indicate that RNNS is efficient in attacking defended models and can be employed for adversarial training.

## 1 Introduction

Recently, since programming language can be seen as one kind of textual data and also inspired by the success of deep learning for text processing and understanding, researchers have tried to pre-train code models such as CodeBERT (Feng et al., 2020), GraphCodeBERT (Guo et al., 2020), Contra-BERT (Liu et al., 2023a) to help developers to solve multiple programming tasks, e.g., code search (Gu et al., 2018; Liu et al., 2023b), code clone detection (White et al., 2016; Li et al., 2017), code summarization (Ahmad et al., 2020; Liu et al., 2020), and vulnerability detection (Zhou et al., 2019). Although these code models have achieved good performance on many code tasks, they are still suffering from robustness issues. A few adversarial attack methods have emerged to evaluate and improve the robustness of code models.

There are certain considerations to be made. Firstly, code pre-training models are frequently deployed remotely, which limits access to the model parameters and renders white-box attacks infeasible. Secondly, among the numerous code-equivalent transformation methods, variable substitution exerts the most significant influence on the resilience of large code models while being the least detectable transformation (Li et al., 2022). As a result, black-box attack techniques based on variable substitution have emerged as a valuable avenue for research and multiple works have been proposed such as ALERT (Yang et al., 2022) and MHM (Zhang et al., 2020).

However, these works have three limitations: 1) The number of substitute variables is limited and lacks diversity, which lowers the upper bound of the attack success rate. For example, ALERT employs 60 substitute variables for each variable, which are generated by a pre-trained model, and the substitute variables lack diversity. MHM also randomly selects 1500 words from a fixed dictionary as substitute variables. 2) The verification cost of substitute variables is high. To verify the attack effect of each substitute, it is necessary to replace the source variable with an adversarial sample and perform an actual attack on the victim model. ALERT uses a traversal method to select substitute variables, and in order to reduce the number of attacks, it limits the number of substitute variables; MHM uses a random sampling method to select substitute variables in order to reduce the number of attacks. Neither method is conducive to cost-effective attacks. 3) The generated adversarial samples have

---

*clark.zhang@huawei.com
† corresponding author: ma_wei@ntu.edu.sg

large perturbations. Each adversarial sample usually needs to replace multiple original variables to succeed in attacking, and MHM easily generates semantically incoherent and excessively long variable names.

To address the aforementioned challenges, in this paper, we propose a search-based black-box adversarial attack method to create challenging adversarial samples based on the search seed vector in the variable representation space, namely **Representation Nearest Neighbor Search** (RNNS). Specifically, RNNS, first utilizes publicly available real code datasets to construct a large original substitute set, denoted as $subs_{original}$. Then, based on the previous attack results, RNNS predicts the search seed vector required for the next round of attacks and efficiently searches for the $k$ nearest substitutes to the seed vector from the large-scale original substitute set to form the $subs_{topk}$, where $k$ is much smaller than the size of the original substitute set. The generation process of the $subs_{topk}$ does not involve attacking the victim model even once. Furthermore, the length and similarity of the substitute must adhere to specific perturbation constraints to prevent excessive deviations from $var$.

To evaluate the effectiveness of RNNS, we investigate three pre-trained code models, Code-BERT (Feng et al., 2020), GraphCodeBERT (Guo et al., 2020) and CodeT5 (Wang et al., 2021), and perform the attack on six code tasks in three programming languages, i.e., Java, Python, and C. The results on 18 victim models demonstrate that compared to the approaches MHM and ALERT, RNNS achieves a higher attack success rate (ASR) with a maximum of about 100% improvement and 18/18 times as the winner. Meanwhile, RNNS needs fewer query times (QT) with 8/18 times as the winners. Furthermore, we analyze the quality of adversarial examples statistically and find that RNNS introduces minor perturbations. In the end, we apply RNNS to attack three defended models and find that our approach outperforms the baselines by up to 32.07% ASR. We also use adversarial examples to improve the model's robustness through contrastive adversarial training.

## 2 Preliminaries

### 2.1 Textual Code Processing

The nature of code data (in text format with discrete input space) makes it impossible to feed one

Figure 1: One code model demo on the downstream task.

code input $x$ directly into deep learning models. Thus, transferring code data to learnable continuous vectors is the first step in source code learning. **Dense encoding** (Zhelezniak et al., 2020) is one common method used to vectorize textual code data. To do so, first, we need to learn a tokenizer that splits the code text into a token sequence which is called **Tokenization**. After tokenization, code $x$ is represented by a sequence of tokens, namely, $x = (s_0, ..., s_j, .., s_l)$ where $s_i$ is one token. Then, the code vocabulary dictionary is built by using all the appeared tokens $s_i$, denoted $\mathbb{V}$. After that, every word (token) in $\mathbb{V}$ is embedded by learned vectors $\boldsymbol{v}_i$ with dimension $d$. Here, we use $\boldsymbol{E}^{|\mathbb{V}| \times d}$ to represent the embedding matrix for $\mathbb{V}$. Finally, $x$ can be converted into a embedding matrix $\boldsymbol{R}^{l \times d} = (\boldsymbol{v}_0, ..., \boldsymbol{v}_j, .., \boldsymbol{v}_l)$. After this code encoding, pre-trained code models based on the transformer take the matrix $\boldsymbol{R}^{l \times d}$ as inputs and learn the contextual representation of $x$ for downstream tasks via pre-training such as Masked Language Modeling (MLM) and Causal Language Modeling (CLM).

Figure 1 illustrates the main steps of the code processing models for the downstream classification tasks. First, we tokenize the textual code $x$ into a token sequence that is represented in a discrete integer space. Then, we map the discrete sequence ids into the token vector space $R^{l \times d}$. Next, we feed the token vectors into the task model $f(\theta)$. $f(\theta)$ is built on top of pre-trained models. Finally, we can predict the domain probabilities after fine-tuning.

### 2.2 Problem Statement

Since many critical code tasks are classification problems, e.g., defect prediction and code clone detection. In this paper, we focus on the adversarial attack for code classification tasks. Considering a code classification task, we use $f(x; \theta) \rightarrow y : \boldsymbol{R}^{l \times d} \rightarrow \mathbb{C} = \{i | 0 \leq i \leq n\}$ to denote the victim model that maps a code token sequence $x$ to a label $y$ from a label set $\mathbb{C}$ with size $n$, where

$l$ is the sequence length and $d$ is the token vector dimension, and $i$ is one integer. By querying dictionary dense embedding $\boldsymbol{E}^{|\mathbb{V} \times d|}$, a code token sequence $x = (s_0, ..., s_j, .., s_l)$, is vectorized into $\boldsymbol{R}^{l \times d}$. Adversarial attacks for code models create an adversarial example $x'$ by modifying some vulnerable tokens of $x$ with a limited maximum perturbation $\epsilon$ to change the correct label $y$ to a wrong label $y'$. Simply, we get a perturbed $x'$ by modifying some tokens in $(s_0, ..., s_j, .., s_l)$ such that $f(x'; \theta) \neq f(x; \theta)$ where $x' = x + \sigma$ and $x'$ has to have the same behavior with $x$, $+$ represent perturbation execution, $\sigma$ is the perturbation code transformation for $(s_0, ..., s_j, .., s_l)$, and $\sigma \leq \epsilon$. We target the more practical attacking scenario – black-box attack that requires less information. We assume we cannot access the model parameters and can only utilize the final output of model $f(x; \theta)$ to conduct the attack.

## 3 Methodology

### 3.1 Motivation

As mentioned in the introduction, the current methods face three limitations: 1) there is a limited number of substitute variables; 2) there is a high verification cost associated with substitute variables; and 3) the generated adversarial samples often exhibit large perturbations. Among these limitations, the second one holds the utmost significance as it significantly impacts both the first and third limitations. Due to the high cost involved, it becomes challenging to generate diverse adversarial examples within a reasonable budget. Additionally, attackers tend to introduce large perturbations without employing any perturbation constraints in order to maximize their attacks.

To address these limitations, the **first** question arises: "*Could we substantially reduce the verification cost while allowing for unrestricted diversity of substitute variables and minimizing perturbations?*" To delve into the reasons behind the second limitation, we need to analyze its underlying factors. The low verification efficiency of the substitute set stems from the fact that each substitute can only be verified by constructing an adversarial sample to replace the original variable and then launching an actual attack on the victim model. This realization leads to the **second** question: "*Is it feasible to predict the attack effect of a substitute instead of constructing an adversarial sample to attack the victim model?*"

---

**Algorithm 1:** RNNS

**Hyperarameter:** maximum attacking iteration $max\_itr$
**Input:** input code $x$ with ground label $y$, original substitute set $subs_{original}$
**Output:** adversarial example $x'$, attacking result $is\_suc$

1   $x' = x$
2   $prob_{min} = 1.0$
3   $vars = ExtractVar(x)$
4   $vars = RankVarsWithUncertainty(vars)$
5   **for** $var \in vars$ **do**
6     $sub_{pre} = var$
7     $sub_{cur} = var$
8     $\Delta e_{smo} = 0$
9     $i = 0$
10    $is\_suc = false$
11    **while** $i < max\_itr$ **do**
12      $e_{seed}, \Delta e_{smo} = PredictSeed(sub_{pre}, sub_{cur}, \Delta e_{smo})$
13      $subs_{topk} = SearchTopkSub(e_{seed}, subs_{original}, var)$
14      **for** $sub \in subs_{topk}$ **do**
15       $x'_{tmp} = Replace(x', sub_{cur}, sub)$
16       $prob_y, y' = f(x'_{tmp}; \theta)$
17       **if** $prob_y < prob_{min}$ **then**
18        $x' = x'_{tmp}$
19        $sub_{pre} = sub_{cur}$
20        $sub_{cur} = sub$
21        $prob_{min} = prob_y$
22       **end**
23       **if** $y! = y'$ **then**
24        $is\_suc = true$
25        return $x', is\_suc$
26       **end**
27      **end**
28     **end**
29    **end**
30   return $x', is\_suc$

---

Given input code $x$ and one of its variables $var$, different substitutes can be used to replace it to obtain different adversarial samples. After attacking the victim model, the probability of the label will also change. Conversely, if we want to reduce the probability of this label, the **third** question is following, "*how to choose relatively better substitutes that can reduce the model confidence from a large-scale original substitute set?*". It is possible to select good substitutes without actual attack if we can forecast, which is implemented by RNNS.

The core idea of RNNS is maintaining a search seed updated based on the historical attack. The search seed is employed to search next adversarial substitutes that are possible to attack successfully. Since substitutes are discrete and cannot be directly involved in calculations, we first use a variable name pre-trained encoder denoted as $E$ to map substitutes to a unified continuous representation vector space. Then, based on the representation vectors of substitutes that have participated in the attack, we predict the search seed vector $e_{seed}$ for the next round of the substitute selection. Finally, we calculate the similarity between $e_{seed}$ and the representation vector of substitutes and then select relatively better substitutes. For specific details, please refer to Section 3.2.3.

### 3.2 Representation Nearest Neighbor Search

Algorithm 1 shows the workflow of our approach, First, we collect the original substitute set from public real code, following the process described

in Section 3.2.1. We extract variables from the input code and sort them according to their uncertainty, referring to Section 3.2.2 (Line 3-4). We replace variables in sequence to form attack samples (Line 5). For a given $var$, we first initialize the optimal substitute for this current iteration $sub_{cur}$ and the optimal substitute for the previous iteration $sub_{pre}$ to the $var$. Then, we initialize the accumulated smooth increment of the representation vector $\Delta e_{smo}$ to a zero vector. $\Delta e_{smo}$ is used to record the historical representation change of the search seed $e_{seed}$. We now commence the iterative attack process, as delineated in Line 11. We predict the search seed vector $e_{seed}$ with the process described in Section 3.2.3 (Line 12), and then extract $topk$ substitutes based on $e_{seed}$ to form the candidate substitutes $subs_{topk}$ with the process described in Section 3.2.4 (Line 13). Subsequently, we replace $sub_{cur}$ in $x'$ with each substitute in $subs_{topk}$ to obtain the corresponding temporary adversarial sample $x'_{tmp}$ (Line 14-15). $x'$ is the current code that we are trying to attack and it is initialized with the original code $x$. We use $x'_{tmp}$ to attack the victim model and obtain the probability $prob_y$ of the ground-truth label y and predicted label $y'$ (Line 16). If the probability of the ground-truth label y hits a new low ($< prob_{min}$), we update $x'$, $sub_{pre}$, $sub_{cur}$ and $prob_{min}$ (Line 17-22). $prob_{min}$ records the minimum probability of label $y$ during the attack process. If $x'_{tmp}$ causes the victim model to predict an incorrect label, this attack is successful and returns the successful adversarial sample (Line 23-26); otherwise, proceed to the next iteration until all variables have completed iteration and return the final adversarial sample and attack result (Line 30).

### 3.2.1 Collecting Large Original Substitute Set

We have developed a tool for variable extraction that leverages the tree-sitter framework[1]. This tool, henceforth denoted as $ExtractVar$ (see Line 3), operates in three distinct steps. In the first step, we extract all variables from the current dataset and then filter out duplicates. During the second step, each valid variable is tokenized, and we compute the embedding for each token using the variable-name encoder $E$ that is pre-trained on CodeSearch-Net[2]. We then apply a mean pooling operation on these tokens to determine the variable's embedding. In the third step, we retain all the chosen variables

along with their associated embeddings as the initial substitute set, represented as $subs_{original}$.

### 3.2.2 Computing Uncertainty

Given a specific code $x$, we replace each instance of $var \in x$ with a set of predefined fixed variables $VarArray$, resulting in a set of mutated codes denoted as $X_{var}^{mutated}$. These mutated codes are subsequently utilized to query the victim model, allowing us to obtain the probability distribution for each class. A greater variance in the distribution signifies increased uncertainty for $var$, suggesting that $var$ should be prioritized for replacement. The uncertainty associated with $var$ is defined as follows:

$$uncertainty_{var} = \frac{1}{C}\sum_{i=1}^{C} variance(P_{var}^i)$$

, where $P_{var}^i = \{p_{var}^i(x)|\forall x \in X_{var}^{mutated}\}$, $C$ is the number of labels, $p_{var}^i(x)$ is the model probability for label $i$ given the mutated code $x$, and $variance$ denotes the standard variance function. A larger and more diverse $X_{var}^{mutated}$ ensures a closer approximation of $uncertainty_{var}$ to the true value. It is important to note, however, that the magnitude of the change length must not be excessively large, as this would result in all probability changes converging to a single point. This is because samples subjected to large changes deviate significantly from the original, leading to a substantial decrease in the model confidence levels. Subsequently, we arrange the variables in descending order based on their uncertainties. The greater the uncertainty of a variable, the more valuable it is for attack. This process is denoted as $RankVarsWithUncertainty$ at line 4. In our implementation, the size of this variable array $VarArray$ is 16, and the variable length ranges from 1 to 5.

### 3.2.3 Predicting Search Seed

To filter out superior substitutes from the substantial $subs_{original}$, it becomes necessary to predict the search seed within the substitute representation vector space. Given the optimal substitute $sub_{cur}$ of the current round, the optimal substitute $sub_{pre}$ from the previous round, and the accumulated smooth increment of the representation vector, denoted as $\Delta e_{smo}$, from all preceding rounds of iteration, we initially compute the increment of the representation vector in the current round, $\Delta e$:

$$\Delta e = E(sub_{cur}) - E(sub_{pre})$$

[1] https://tree-sitter.github.io/tree-sitter

[2] https://huggingface.co/datasets/code_search_net

| Task | Train / Val / Test | CodeBERT | GraphCodeBERT | CodeT5 |
|------|--------------------|----------|---------------|--------|
| Defect | 21,854 / 2,732 / 2,732 | 63.76 | 63.65 | 67.02 |
| Clone | 90,102 / 4,000 / 4,000 | 96.97 | 97.36 | 97.84 |
| Authorship | 528 / – / 132 | 82.57 | 77.27 | 88.63 |
| C1000 | 320,000 / 80,000 / 100,000 | 82.53 | 83.79 | 84.46 |
| Python800 | 153,600 / 38,400 / 48,000 | 96.39 | 96.29 | 96.79 |
| Java250 | 48,000 / 11,909 / 15,000 | 96.91 | 97.27 | 97.72 |

Table 1: Datasets and Victim Model Performance (Accuracy, %).

, where $E$ is variable name encoder, trained on CodeSearchNet by masked language modelling independently so that RNNS is independent of victim downstream-task models. Then we update the $\Delta e_{smo}$,

$$\Delta e_{smo} = (1 - \alpha)\Delta e_{smo} + \alpha \Delta e$$

, where $\alpha$ is a smooth rate limited 0 to 1, Finally, we predict the search seed $e_{seed}$:

$$e_{seed} = E(sub_{cur}) + \Delta e_{smo}$$

This process is denoted as $PredictSeed$ at line 12.

### 3.2.4 Searching Top-K Substitutes

Initially, we filter out substitutes from $subs_{original}$ that comply with two constraints: 1) $1 - sim(E(sub), E(var)) < \epsilon$ and 2) $|len(sub) - len(var)| < \delta$, where $var$ refers to the original variable in the input code that is to be replaced, $sim(.)$ is the similarity calculation function. $E(.)$ is the variable name encoder, and $len(.)$ is used to calculate the length of the variable name. Then, we calculate the similarity between the search seed $e_{seed}$ and the substitutes that are filtered by the two constraints and select the $k$ most similar substitutes to form $subs_{topk}$. This process is denoted as $SearchTopkSub$ at line 13. In our experiment, $\epsilon = 0.15$, $\delta = 4$, $k = 60$, $sim(.)$ is cosine similarity.

## 4 Experimental Setup

**Dataset and Model.** To study the effectiveness and efficiency of RNNS, we conduct experiments on three popular programming languages (C, Python, and Java). For the datasets, we employed six widely studied open-source datasets that cover four important code tasks. Specifically, Big-CloneBench (Wang et al., 2020) is one code clone detection dataset named Clone. Devign (Zhou et al., 2019) is a dataset used for vulnerability detection, named Defect. For authorship prediction, we use the dataset provided by (Alsulami et al., 2017).

Besides, we utilize three problem-solving classification tasks, Java250, Python800, and C1000, provided by ProjectCodeNet (Puri et al., 2021). For all the datasets (except for authorship prediction which does not have enough data samples), we follow the original papers to split the data into the training set, validation set, and test set. Authorship prediction only has two split parts, training data and test data.

For the code models, we follow the previous work (Yang et al., 2022) and investigate two pre-trained models CodeBERT (Feng et al., 2020), and GraphCodeBERT (Guo et al., 2020). Besides, we add one more powerful model CodeT5 (Wang et al., 2021) in our study. Table 1 summarizes the details of our employed datasets and fine-tuned models.

**Evaluation Metric.** To evaluate the effectiveness of adversarial attack methods, we employ the commonly used attack success rate (ASR) (Yang et al., 2022) as the measurement. To evaluate the efficiency of the attack methods, we use query times (QT) to check the average number of querying the victim model for one input code. Finally, we use the change of replaced-variable length and the number of replaced variables to study the quality/perturbation of adversarial examples. A smaller score means the attack method can generate adversarial examples with less perturbation injection.

**Baseline.** We compare RNNS with two black-box attack baselines, MHM (Zhang et al., 2020) and NaturalAttack (ALERT) (Yang et al., 2022). MHM is a sampling search-based black-box attack that generates the substitutes from the vocabulary based on lexical rules for identifiers. MHM employs synthesized tokens as the candidates of substitutes, which could introduce meaningless variable names. ALERT is a recently proposed attack method that combines greedy attack and genetic algorithm to find the substitutes. We also use two textual attack algorithms PSO (Zang et al., 2020) and LSH (Maheshwary et al., 2021) as minor baselines, since they are not designed for code models.

**Implementation.** We implement our approach in PyTorch and run all experiments on 32G-v100 GPUs. We reuse the source code from the baselines. We make our implementation [3] publicly available.

| Task+Model | ALERT | | MHM | | RNNS | |
|---|---|---|---|---|---|---|
| | ASR | QT | ASR | QT | ASR | QT |
| Clone+CodeBert | 28.67 | 2155.39 | 39.66 | 972.15 | **46.50** | **666.48** |
| Clone+GraphCodeBert | 10.40 | 1466.68 | 9.58 | **490.99** | **41.28** | 1122.01 |
| Clone+CodeT5 | 29.20 | 2359.70 | 38.79 | 1069.06 | **39.61** | **895.79** |
| Defect+CodeBert | 52.29 | 1079.68 | 50.51 | 862.18 | **69.18** | **588.35** |
| Defect+GraphCodeBert | 74.29 | 621.77 | 75.19 | 539.93 | **81.63** | **404.73** |
| Defect+CodeT5 | 76.66 | 721.02 | 86.51 | **344.08** | **89.45** | 344.29 |
| Authorship+CodeBert | 34.98 | **682.57** | 64.70 | 775.11 | **73.39** | 1029.59 |
| Authorship+GraphCodeBert | 58.82 | 1227.36 | 75.49 | **632.10** | **80.39** | 696.64 |
| Authorship+CodeT5 | 64.95 | 1078.40 | 66.97 | **715.89** | **71.79** | 970.44 |
| Java250+CodeBert | 50.50 | 958.96 | 74.03 | 961.60 | **75.12** | **815.91** |
| Java250+GraphCodeBert | 46.74 | 1026.15 | 46.05 | 946.52 | **72.30** | **853.74** |
| Java250+CodeT5 | 52.04 | 1189.42 | 30.59 | 1107.95 | **63.80** | **1049.46** |
| Python800+CodeBert | 58.30 | **513.63** | 56.67 | 919.37 | **77.88** | 514.19 |
| Python800+GraphCodeBert | 51.87 | **577.70** | 54.15 | 917.92 | **71.42** | 730.14 |
| Python800+CodeT5 | 52.84 | 777.20 | 36.95 | 1127.44 | **69.07** | **662.28** |
| C1000+CodeBert | 53.50 | 525.43 | 59.75 | **340.88** | **72.96** | 537.76 |
| C1000+GraphCodeBert | 52.68 | **566.18** | 45.93 | 837.09 | **72.23** | 634.27 |
| C1000+CodeT5 | 47.86 | 843.33 | 36.45 | **668.15** | **59.00** | 697.06 |
| Count | 0/18 | 4/18 | 0/18 | 6/18 | 18/18 | 8/18 |

Table 2: Comparison results with MHM, and ALERT, ASR %. Count: the number of best results achieved.

## 5 Results Analysis

### 5.1 Attack Effectiveness and Efficiency

We compare RNNS with two methods MHM (Zhang et al., 2020) and NaturalAttack (ALERT) (Yang et al., 2022) on six datasets and 18 victim models that have been fine-tuned for the downstream tasks. Table 2 shows the comparison results where the last row *Count* indicates how many times this method achieves the best results. We can see that RNNS achieves the best performance for 18/18 times in terms of ASR, and the lowest cost for 8/18 times in terms of QT in Table 2. Both of the indicators are better than the baselines. The two baselines have zero best ASR for all victim models and all datasets. The lowest QTs achieved by ALERT and MHM are 4 and 6, respectively. We conclude that for effectiveness and efficiency, RNNS outperforms ALERT and MHM in all cases. Especially, MHM and ALERT fail to attack GraphCodeBERT on BigClone dataset, and only have $9.58\%$ and $10.4\%$ ASR respectively, while RNNS has more than $40\%$ ASR. RNNS has almost two times larger ASR than MHM on Java250+CodeT5 and Python800+CodeT5.

It should be noted that high ASR is not due to large QT. As shown in Table 2, the three groups of experiments with the most QTs are Clone+GraphCodeBert, Java250+CodeT5, and Authorship+CodeBert, with ASRs of $41.28\%$, $63.80\%$, and $73.39\%$, respectively, which are not the highest. On the contrary, Defect+CodeT5 has the highest

[3]https://github.com/18682922316/RNNS-for-code-attack

ASR of 89.45%, but QT is the smallest. Therefore, there is no absolute causal relationship between QT and ASR.

### 5.2 Perturbation of Adversarial Example

We conduct a study about the quality of the adversarial examples to check if RNNS can generate looking-normal code, e.g., avoiding naively increasing the variable name length. To do so, firstly, we count the average length of the original variable and adversarial variables as demonstrated by Table 3. We also compute the mean and variances of their difference. Besides, we compute the average number of the replaced variables for the successful attack as shown in Table 4. Low values mean the inputs are modified less, and high qualities.

In Table 3, the 2nd, 5th, and 8th columns are the average length for original variables (named *Var Len*) that are replaced. The 3rd, 6th, and 9th columns are the average lengths for adversarial variables (named *Adv Var Len*). The 4th, 7th, and 10th columns are the average and variance ($mean \pm variance$) of the absolute length difference between original variables and adversarial variables (named *Difference*). We observe that MHM prefers to replace the long-length variables while RNNS likes replacing short-length variables if we compare the 2nd and 5th columns. Meanwhile, the change of variable length from RNNS is less than MHM. MHM introduces the average length difference of 3.39-6.82 while RNNS only has 2.02-2.54. MHM has much higher variances than RNNS in the length change. ALERT uses shorter adversarial variable names than RNNS

| Task+Model | RNNS | | | MHM | | | ALERT | | |
|---|---|---|---|---|---|---|---|---|---|
| | Var Len | Adv Var Len | Difference | Var Len | Adv Var Len | Difference | Var Len | Adv Var Len | Difference |
| Clone+CodeBert | 6.12 | 6.79 | 2.35 ± 4.50 | **6.47** | **10.6** | 6.34 ± 10.98 | 5.91 | 6.21 | 1.32 ± 2.02 |
| Clone+GraphCodeBert | 6.32 | 6.97 | 2.54 ± 6.43 | **6.58** | **10.41** | 6.82 ± 21.67 | 5.50 | 5.93 | 1.45 ± 2.49 |
| Clone+CodeT5 | 6.45 | 6.69 | 2.51 ± 8.30 | **6.46** | **10.46** | 6.17 ± 25.78 | 6.25 | 6.61 | 1.32 ± 2.72 |
| Defect+CodeBert | 4.64 | 5.44 | 2.08 ± 2.49 | 4.44 | **9.59** | 6.57 ± 28.78 | **4.85** | 5.06 | 1.36 ± 1.93 |
| Defect+GraphCodeBert | 4.08 | 5.34 | 2.13 ± 1.83 | 4.37 | **9.73** | 6.48 ± 26.51 | **4.47** | 5.22 | 1.33 ± 1.83 |
| Defect+CodeT5 | 3.95 | 5.17 | 2.03 ± 1.93 | 4.33 | **9.81** | 6.59 ± 29.98 | **4.36** | 5.01 | 1.27 ± 1.57 |
| Authorship+CodeBert | 3.81 | 5.18 | 2.28 ± 1.56 | 3.97 | **7.94** | 5.45 ± 16.72 | **4.42** | 5.35 | 1.40 ± 2.25 |
| Authorship+GraphCodeBert | 3.69 | 5.23 | 2.36 ± 1.71 | **4.39** | **7.64** | 5.24 ± 15.38 | 3.74 | 4.46 | 1.22 ± 1.82 |
| Authorship+CodeT5 | **3.95** | 5.18 | 2.03 ± 2.66 | **3.95** | **7.98** | 5.59 ± 20.94 | 3.81 | 4.50 | 1.22 ± 1.62 |
| Java250+CodeBert | 2.35 | 4.22 | 2.11 ± 1.02 | 3.21 | **6.50** | 4.34 ± 15.20 | **3.22** | 3.65 | 0.94 ± 1.63 |
| Java250+GraphCodeBert | 2.48 | 4.31 | 2.13 ± 1.07 | **3.13** | **6.59** | 4.42 ± 14.84 | 3.05 | 3.50 | 0.98 ± 1.54 |
| Java250+CodeT5 | 2.76 | 4.47 | 2.10 ± 1.17 | **3.20** | 6.54 | 4.33 ± 14.60 | 3.16 | **7.31** | 4.41 ± 18.73 |
| Python800+CodeBert | 1.50 | 3.54 | 2.21 ± 1.02 | **1.97** | **5.11** | 3.64 ± 9.06 | 1.78 | 2.27 | 0.64 ± 1.34 |
| Python800+GraphCodeBert | 1.88 | 3.90 | 2.18 ± 0.78 | **1.99** | **6.01** | 4.46 ± 16.52 | 1.80 | 2.33 | 0.76 ± 1.30 |
| Python800+CodeT5 | 1.65 | 3.59 | 2.13 ± 0.95 | **1.97** | 4.95 | 3.49 ± 8.18 | 1.88 | **5.84** | 4.10 ± 12.64 |
| C1000+CodeBert | 1.58 | 3.44 | 2.08 ± 0.88 | **2.41** | 5.05 | 3.65 ± 12.02 | 2.13 | 2.52 | 0.67 ± 1.17 |
| C1000+GraphCodeBert | 1.60 | 3.59 | 2.10 ± 0.85 | **2.39** | 5.35 | 3.90 ± 12.98 | 2.18 | 2.67 | 0.66 ± 1.23 |
| C1000+CodeBert | 1.38 | 3.33 | 2.02 ± 0.85 | **2.36** | 4.82 | 3.39 ± 10.98 | 2.10 | **6.56** | 4.74 ± 13.24 |

Table 3: Replaced-variable length comparison, $mean \pm variance$.

with less change because it uses the pre-trained model to generate the replacements that are close to the replaced variables.

Table 4 statistically shows the number of replaced variables. It can be seen that RNNS replaces around an average of 3.6 variables with a smaller variance of around (3.4-4.6) while MHM needs to modify about an average of 5.4 variables with a larger variance ($\geq 11.14$). ALERT also replaces more variables to attack models than RNNS and MHM. RNNS introduces less or equal perturbation than the baselines in terms of length change and change number.

Figure 2 shows one example of RNNS, MHM, and ALERT attack successfully from the Java250 dataset. The changes are highlighted by shadow markers. RNNS only renames one variable **b** to **h**, ALERT renames two variables, while MHM almost renames all variables and also prefers longer names.

## 5.3 Ablation Study

We remove the two search constraints in Section 3.2.4, denoted this variant of RNNS as RNNS-Unlimited. Table 5 shows the comparing results between RNNS-Unlimited and RNNS. RNNS-Unlimited gets the first place for all the tasks in terms of ASR. ASR can be improved by a maximum of 8.35% and a minimum of about 2% after removing limitations. For QT, RNNS-Unlimited only loses 3 times among 18 evaluations. The improvement of RNNS-Unlimited is not surprising with respect to ASR and QT. Because RNNS-Unlimited can search the adversarial examples in the non-similar real names and use very long variable names.

## 5.4 Attack Defended Model and Retraining

**Attack Defended Model.** We employ RNNS and MHM to attack the three defended models provided by ALERT (Yang et al., 2022). These models are prepared by adversarial fine-tuning. Table 6 presents the results. We can see that RNNS outperforms MHM in two tasks, and MHM is better in one task. This experiment setting actually is not friendly for RNNS because ALERT (Yang et al., 2022) uses the replacements from pre-trained models which implicitly have the semantic constraint.

**Retraining.** We use the adversarial examples from RNNS to retrain the victim models of CodeBERT by contrastive adversarial learning. We use three 3 datasets, Defect, Authorship, and Java250. We generate the adversarial examples on the whole training dataset for them. Table 7 presents the results, all approaches achieve much lower ASR compared with the previous. RNNS adversarial examples can improve the mode robustness through contrastive adversarial retraining. If we compare Defect/Authorship+CodeBERT in Table 7 and Table 6, we can find that both retrained models via RNNS are more robust than the models from ALERT since they have much lower ASRs.

## 5.5 RNNS vs Textual Attack Methods

To compare the effects of RNNS and textual attack methods, We conducted attack experiments on three datasets using the PSO (Zang et al., 2020) and LSH (Maheshwary et al., 2021). The three datasets Defect, Authorship, and Java250, represent three languages respectively, C, Python, and Java. To be fair, the search space of the PSO and LSH is the same as that of RNNS.

As shown in Table 8, the QT of PSO algorithm

| Task | CodeBERT | | | GraphCodeBERT | | | CodeT5 | | |
|---|---|---|---|---|---|---|---|---|---|
| | RNNS | MHM | ALERT | RNNS | MHM | ALERT | RNNS | MHM | ALERT |
| Clone | 3.55 ± 4.60 | 6.72 ± 16.57 | 6.86 ± 18.85 | 4.12 ± 4.94 | 6.21 ± 15.13 | 6.95 ± 18.99 | 3.43 ± 5.00 | 5.68 ± 14.01 | 7.65 ± 25.57 |
| Defect | 3.39 ± 4.96 | 2.78 ± 7.89 | 3.49 ± 3.99 | 2.67 ± 1.75 | 2.84 ± 9.50 | 4.10 ± 11.05 | 2.51 ± 1.45 | 2.16 ± 3.58 | 3.49 ± 3.99 |
| Authorship | 4.24 ± 7.47 | 7.52 ± 25.82 | 6.60 ± 22.96 | 3.65 ± 3.32 | 6.67 ± 22.29 | 7.75 ± 33.12 | 4.39 ± 9.00 | 5.72 ± 13.02 | 6.06 ± 18.74 |
| Java250 | 3.87 ± 4.70 | 7.11 ± 21.18 | 7.82 ± 28.96 | 3.87 ± 4.25 | 6.41 ± 16.24 | 7.83 ± 25.06 | 4.71 ± 6.87 | 7.04 ± 15.29 | 8.92 ± 25.97 |
| Python800 | 3.06 ± 1.87 | 5.21 ± 12.28 | 4.96 ± 8.47 | 4.12 ± 3.68 | 5.00 ± 10.83 | 4.63 ± 6.76 | 3.57 ± 3.04 | 5.29 ± 13.51 | 6.18 ± 11.45 |
| C1000 | 3.00 ± 1.86 | 4.42 ± 7.49 | 4.13 ± 5.59 | 3.37 ± 2.38 | 5.14 ± 7.30 | 4.88 ± 6.24 | 3.39 ± 2.48 | 5.20 ± 7.43 | 5.43 ± 6.99 |
| mean | **3.52 ± 4.24** | 5.63 ± 15.21 | 5.65 ± 14.80 | **3.63 ± 3.39** | 5.38 ± 13.55 | 6.02 ± 16.87 | **3.67 ± 4.64** | 5.18 ± 11.14 | 6.29 ± 15.45 |

Table 4: Replaced-variable number comparison, $mean \pm variance$

Figure 2: Case study. Original vs. RNNS vs. MHM vs. ALERT

| Task | CodeBERT | | | | GraphCodeBERT | | | | CodeT5 | | | |
|---|---|---|---|---|---|---|---|---|---|---|---|---|
| | RNNS-Unlimited | | RNNS | | RNNS-Unlimited | | RNNS | | RNNS-Unlimited | | RNNS | |
| | ASR | QT | ASR | QT | ASR | QT | ASR | QT | ASR | QT | ASR | QT |
| Defect | **72.29** | 590.98 | 69.18 | **588.35** | **87.77** | **381.82** | 81.63 | 404.73 | **91.64** | **338.41** | 89.45 | 344.29 |
| Clone | **50.66** | 955.97 | 46.50 | **666.48** | **48.16** | 1105.11 | 41.28 | 1122.01 | **41.38** | 920.65 | 39.61 | **895.79** |
| Authorship | **91.74** | 447.68 | 73.39 | 1029.59 | **91.17** | 438.69 | 80.39 | 696.64 | **88.88** | 620.56 | 71.79 | 970.44 |
| C1000 | **74.70** | 502.02 | 72.96 | 537.76 | **76.82** | 498.64 | 72.23 | 634.27 | **61.96** | 704.95 | 59.00 | 697.06 |
| Python800 | **83.90** | 460.92 | 77.88 | 514.19 | **79.00** | 496.30 | 71.42 | 730.14 | **72.69** | 646.59 | 69.07 | 662.28 |
| Java250 | **79.70** | 760.97 | 75.12 | 815.91 | **81.94** | 744.57 | 72.30 | 853.74 | **75.52** | 910.97 | 63.80 | 1049.46 |
| Count | 6/6 | 4/6 | 0/6 | 2/6 | 6/6 | 6/6 | 0/6 | 0/6 | 6/6 | 5/6 | 0/6 | 1/6 |

Table 5: Results of ablation study, before and after removing constraints, ASR %.

| Defended Model | RNNS | | MHM | |
|---|---|---|---|---|
| | ASR | QT | ASR | QT |
| Clone+CodeBert | 12.90 | **958.35** | 28.17 | 1245.75 |
| Defect+CodeBert | 95.37 | 282.20 | 92.23 | 283.66 |
| Authorship+CodeBert | 51.88 | 1524.40 | 43.26 | **1026.08** |

Table 6: Attack defended models, ASR %.

| | ACC | ASR(RNNS) | ASR(MHM) | ASR(ALERT) |
|---|---|---|---|---|
| Authorship | 90.62 | 19.81 | **23.58** | 14.28 |
| Defect | 65.14 | **40.46** | 23.69 | 24.53 |
| Java250 | 97.63 | 19.67 | 6.65 | **42.91** |

Table 7: Results of contrastive adversarial retraining, model: CodeBERT.

| Task+Model | RNNS | | PSO | | LSH | |
|---|---|---|---|---|---|---|
| | ASR | QT | ASR | QT | ASR | QT |
| Defect+CodeBert | **69.18** | 588.35 | 63.63 | 3945.04 | 26.62 | **321.78** |
| Authorship+CodeBert | **73.39** | 1029.59 | 52.29 | 4350.00 | 19.26 | **458.55** |
| Java250+CodeBert | **75.12** | 815.91 | 47.3 | 5076.02 | 31.58 | **397.05** |

Table 8: RNNS vs PSO and LSH, ASR %.

of word replacement on natural language semantics.

RNNS's QT is 1.8-2.2 times that of LSH, and the QT has dropped significantly. However, LSH's ASR is inferior to RNNS by 42.56%-54.13%. For code variable attacks, LSH has high efficiency, but its effectiveness is relatively low. One possible reason for LSH causing low ASR is the distribution of adversarial samples in each bucket is uneven.

# 6 Related Work

Adversarial attacks for code models have been widely studied (Yang et al., 2022; Liu et al., 2023a; Li et al., 2023; Jha and Reddy, 2023). These works can be generally categorized into black-box attacks and white-box attacks. A black-box attack for code models queries the model outputs and selects the substitutes using a score function. For example,

is 4.22-6.7 times that of RNNS, and the ASR of PSQ algorithm is 5.55% - 27.82% lower than that of RNNS algorithm. It can be inferred that for code variable attacks, combinatorial optimization is inefficient when the substitute set of variables is relatively large. The main reasons are the following two points. Firstly, code segments are generally longer, and the substitute set of code variables is much larger than the synonym set of natural language words. Secondly, the impact of variable replacement on code semantics is smaller than that

| Algorithm | Substitutes Size | Substitutes Source | Replacement Position | Substitutes Selection |
|-----------|------------------|--------------------|----------------------|-----------------------|
| MHM | medium | vocabulary | random | random sample |
| ALERT | small | model generation | importance score | traverse |
| RNNS | large | real public variables | uncertainty score | efficient constrained search |

Table 9: Difference between RNNS to the others.

ALERT (Yang et al., 2022) finds the adversarial examples using variable-name substitutes generated by pre-trained masked models. MHM (Zhang et al., 2020) uses Metropolis–Hastings to sample the replacement of code identifiers. STRATA (Springer et al., 2020) generates adversarial examples by replacing the code tokens based on the token distribution. Chen et al. (2022) apply pre-defined semantics-preserving code transformations to attack code models. CodeAttack (Jha and Reddy, 2023) uses code structure to generate adversarial data. White-box attacks require the code model gradient to modify inputs for adversarial example generation. CARROT (Zhang et al., 2022) selects code mutated variants based on the model gradient. Henkel et al. (2022) attack code models by gradient-based optimization of the abstract syntax tree transformation. Srikant et al. (2021) uses optimized program obfuscations to modify the code. DAMP (Yefet et al., 2020) derives the desired wrong prediction by changing inputs guided by the model gradient.

Table 9 demonstrates the differences among RNNS, MHM (Zhang et al., 2020) and ALERT (Yang et al., 2022). MHM and ALERT represent the two methodologies most closely aligned with our research. Our approach considers identifier replacements like MHM and ALERT, ensuring that the adversarial example keeps the same semantics as the original one. Our substitute size is scalable and can be substantial, and RNNS searches the possible next adversarial example in the substitute space. In our approach, we locate vulnerable variables based on the uncertainty and search $subs_{topk}$ without building adversarial samples and actual attacks. Our goal is to obtain high ASRs by searching real variable names. MHM has the same goal as ours but synthesizes variable names. ALERT sacrifices ASR to make the variable name readable.

## 7 Conclusion

We propose a novel black-box adversarial search-based attack for variable replacement. RNNS has three main contributions: 1) This work proposes a non-generation search-based black-box attacking method via predicting the attack effect of a substitute. This method can greatly reduce the verification cost of the substitute, remove the restrictions on the size and diversity of the substitute set, and achieve a significant improvement in terms of ASR without increasing QT. 2) This work proposes a simple and efficient method for constructing a substitute set. This method can construct a large-scale, diverse, and real substitute set at low cost. 3) The adversarial examples from RNNS can be used to improve the model robustness.

## 8 Limitations

There are some limitations of RNNS. Firstly, RNNS does not revert to the preceding step to persist with the search upon an increase in the model probability of the ground truth label. While the incorporation of this step may bolster the Attack Success Rate (ASR), it could potentially compromise the Query Time (QT). Secondly, the size and diversity of the substitute set significantly influence RNNS; a minimal and homogeneous set can precipitate a diminished attack success rate. Thirdly, RNNS involves multiple hyperparameters whose values need to be manually set. One of the most important parameters is the moving parameter $\alpha$. The number of attacking iterations $max\_itr$ is also significant. We set $\alpha$ to 0.2 and $max\_itr$ to 6 with some small experimental trials. Fourthly, RNNS currently only targets untargeted attack scenarios, for targeted attacks, ASR will be very low when there are many category labels. For example, when performing targeted attacks on Authorship+Codebert with 66 labels, the ASR can only reach 6.4%. How to migrate to targeted attacks is a direction we need to study in the future.

## Acknowledgment

This work is supported by NRF and the CSA under its National Cybersecurity R&D Programme (NCRP25-P04-TAICeN), NRF and DSO National Laboratories under the AI Singapore Programme (AISG Award No: AISG2-RP-2020-019), and NRF Investigatorship NRF-NRFI06-2020-0001. Any opinions, findings and conclusions or recommendations expressed in this material are those of the author(s) and do not reflect the views of NRF and CSA Singapore.

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
