# OpenReview forum: "A Black-Box Attack on Code Models via Representation Nearest Neighbor Search"
_EMNLP/2023/Conference — EMNLP 2023 Findings_

### Official Review · Reviewer_fokm · 2023-08-01

**Soundness:** 3

**Excitement:**

3: Ambivalent: It has merits (e.g., it reports state-of-the-art results, the idea is nice), but there are key weaknesses (e.g., it describes incremental work), and it can significantly benefit from another round of revision. However, I won't object to accepting it if my co-reviewers champion it.

**Paper Topic And Main Contributions:**

This paper presents an adversarial attack on programming-language models such as codeT5 or codebert. The attack replaces variable names in the input code in order to cause the output label of the attacked model to flip. Variables are selected to replace based on the average variance that their replacement causes in the output probability distribution. Once the variable that causes the most variance is found, candidate variable names are found by searching in a variable name embedding space, learned by a masked language model. The search algorithm is based on nearest neighbors search around a seed embedding, which is the embedding of the current name plus a decaying delta vector calculated from the differences between past embeddings. This appears to be like momentum for iterative search in embedding space.

**Questions For The Authors:**

Question a: In the subsection 3.2.2 "Computing Uncertainty", is a new $X_mutated$ being created for each $var \in x$? In the equation above line 307 and the definition of $P_i$ on line 307, I'm not seeing the reference to a specific $var$ for which the variance is being calculated.

Question b: In section 3.2.4 "Searching Top-K Substitutes" you say that you filter out all variable names that don't have a similarity of less than $\epsilon$. I'm not sure it's stated elsewhere but what is this similarity function and what is $\epsilon$ set to? Do you know how many substitutions it filters out? I've seen previous work on adversarial attacks where their constraints were too tight that no actual search was being performed as there was only one possible token to select.

Question c: Regarding the future work on targeted attacks. That's the case where you want to get the model to predict a certain label, right? Couldn't you change line 17 of algorithm 1 to increase the probability of the target label instead of just decreasing the probability of the current label?

**Reasons To Accept:**

The search algorithm appears novel to me and I like the idea of searching embedding space to find adversarial attacks.

Their algorithm achieves attack success rates better than their competitor attacks and appears to take less time.

**Reasons To Reject:**

I'm not sure how relevant this is to the larger NLP community. Most of the cited work appears in IEEE or ACM conferences as they're about programming languages, not human language. Although this work could be adapted to human language, it would not be trivial. Additionally, a lot of citations are of arxiv papers; without looking them all up I can't tell if they've been reviewed or not.

While the attack success rates go up, I'm not sure query time is the natural metric for attacking black-box systems by API call. Previous work in black-box attacks on classifiers for sentiment analysis and entailment has used the total number of queries required to flip the label and not time. Services available by API usually charge based on number of calls and not compute time.

Some notation is unclear but I think that could be fixed for a camera ready version.


**Reproducibility:**

3: Could reproduce the results with some difficulty. The settings of parameters are underspecified or subjectively determined; the training/evaluation data are not widely available.

**Reviewer Confidence:**

2: Willing to defend my evaluation, but it is fairly likely that I missed some details, didn't understand some central points, or can't be sure about the novelty of the work.

---

> ### Author Rebuttal · Authors · 2023-08-29
>
> Thank you for acknowledging the innovation of our method. Code models represent a highly significant and promising application in NLP with substantial industrial value. By QT~ (query times) here, we refer to the number of times the API is queried, not the duration. We used this term to stay consistent with the baseline's original paper, making comparisons more straightforward. We have carefully read your questions and provide the following answers.
>
> **For Q1**:
>
> Yes, we create a set of mutated codes for each variable in code X.
> RankVarsWithUncertainty implementation consists of five steps:
> 1. Specify a fixed variable array VarArray. In this experiment, the size of the variable array is 16, and the variable length ranges from 1 to 5.
> 2. For a certain variable var in the code, replace it with each variable in VarArray in turn, so that 16 mutated codes are obtained.
> 3. Query the victim model with these 16 mutated codes to obtain the probability distribution of each category.
> 4. Calculate the variance of each category probability distribution.
> 5. Calculate the mean of all category variances as the uncertainty of variable var.
> The greater the uncertainty of a variable, the more valuable it is for attack.
>
> **For Q2**:
>
> When selecting TOPK Substitutes, we use cosine similarity. We construct a candidate set of substitutes with similarity > 0.85 (ϵ=0.15) and length difference less than 4 (δ=4). The initial substitutes for the variables are in the tens of thousands (for example, Java250 is 29, 255). After our similarity and length filtering, there are thousands of carefully selected substitutes left (the average selected candidate set for the Java250 variable is 2, 792). Each iteration then selects the top 60 (k=60) most similar substitutes as TOPK substitutes.
> We will specify the values of each parameter in the revised version.
>
>
> **For Q3**:
>
> Yes, targeted attacks refer to making the model predict a specified label. For future study, we conducted experiments on targeted attacks using the method you mentioned. When there are many category labels, the success rate of attacks is very low. For example, when performing targeted attacks on Authorship+Codebert with 66 labels, the ASR can only reach 6.4%.

---

### Official Review · Reviewer_N9F5 · 2023-08-04

**Soundness:** 3

**Excitement:**

3: Ambivalent: It has merits (e.g., it reports state-of-the-art results, the idea is nice), but there are key weaknesses (e.g., it describes incremental work), and it can significantly benefit from another round of revision. However, I won't object to accepting it if my co-reviewers champion it.

**Paper Topic And Main Contributions:**

- The paper proposes representation nearest neighbor search (RNNS), a novel black-box adversarial attack method on code models.
- RNNS trains an additional encoder which embeds a variable into a continuous representation vector.
- Then, RNNS construct variable susbstitute candidates as nearest neighborhood set in the embedding space.
- RNNS measures the uncertainty of each variable in the code and sequentially perturbs variables from higher uncertainty to lower uncertainty until attack success.
- The empirical results show that RNNS achieve higher attack success rate compared to the baseline methods.

**Questions For The Authors:**

- The baseline methods and the proposed RNNS utilize greedy-based attack algorithm. However, there exist several research that utilize combinatorial optimization to attack the language model [4,5]. Can you provide attack result of RNNS equipped with [4] and [5]? Also, you may further accelerate the attack process using locality sensitive hashing by adapting the protocol of [6].
- I am little confused with threat model. According to the section 3.1, it seems that RNNS utilizes training code dataset to train the encoder model, which requires strong assumption on the attack setting. Is your threat model permit access to the training dataset of the victim code model? If not, please clarify which data you utilized to train the encoder model.
- Can you provide some additional qualitative research in the revised version?


[4] Word-level Textual Adversarial Attacking as Combinatorial Optimization, Zang et al., ACL 2020

[5] Query-Efficient and Scalable Black-Box Adversarial Attacks on Discrete Sequential Data via Bayesian Optimization, Lee et al., ICML 2022

[6] A Strong Baseline for Query Efficient Attacks in a Black Box Setting, Maheshwary et al., EMNLP 2021

**Reasons To Accept:**

- The empirical results show the superiority of the proposed method compared to the baseline method.
- The method is simple and intuitive.
- This attack algorithm of RNNS follows the algorithm of greedy-based word-level adversarial attack research [1,2,3]. If I understood correctly, there are two differences.
 1. In the adversarial attack on code models, tokens corresponding to the same variable should be perturbed into the same variable.
 2. Due to difference in token distributions of code models and other NLP models, adversarial attack method on code models cannot directly use the word (token) substitution method of word-level adversarial attack.

Even though the proposed method is not algorithmic novel, it is important research that suggests base word-substitution method for code models.

[1] Generating Natural Language Adversarial Examples through Probability Weighted Word Saliency, Ren et al., ACL 2019

[2] Is BERT Really Robust? A Strong Baseline for Natural Language Attack on Text Classification and Entailment, Jin et al., AAAI 2020

[3] BERT-ATTACK: Adversarial Attack Against BERT Using BERT, Li et al., EMNLP 2020.



**Reasons To Reject:**

- Please refer to questions.

**Reproducibility:**

3: Could reproduce the results with some difficulty. The settings of parameters are underspecified or subjectively determined; the training/evaluation data are not widely available.

**Reviewer Confidence:**

4: Quite sure. I tried to check the important points carefully. It's unlikely, though conceivable, that I missed something that should affect my ratings.

---

> ### Author Rebuttal · Authors · 2023-08-29
>
> Thank you for your feedback. Our attack algorithm is consistent with the idea of greedy strategy, attacking in the most likely direction. Our attack algorithm is based on the word-substitution method in the code model scenario. We have carefully read your questions and provided references.
>
> **For Q1**:
>
> We conducted attack experiments on three datasets using the PSO[4] algorithm and LSH[6] algorithm . The three datasets, Defect, Authorship and Java250, represent three languages respectively, C, Python and Java. To be fair, the search space of the algorithm is the same as that of RNNS. The experimental results are shown in the following table:
>
>
> |     　                     |      RNNS    |   RNNS   |      PSO     |    PSO   |      LSH     |    LSH   |
> |----------------------------|:------------:|:---------------:|:------------:|:---------------:|:------------:|:-------------:|
> |     　                     |     ASR      |     QT          |     ASR      |     QT          |     ASR      |     QT        |
> |     Defect+CodeBert        |     **69.18**    |     588.35      |     63.63    |     3945.04     |     26.62    |    **321.78**    |
> |     Authorship+CodeBert    |     **73.39**    |     1029.59     |     52.29    |     4350.00     |     19.26    |     **458.55**    |
> |     Java250+CodeBert          |     **75.12**    |     815.91      |     47.3     |     5076.02     |     31.58    |     **397.05**    |
>
>
>
>
> 1.	PSO's pop_size is set to 10, and max_iters is 6. As shown in the table above, the QT of the PSO algorithm is 4.22-6.7 times that of RNNS, and the ASR index of the PSQ algorithm is 5.55% - 27.82% lower than that of the RNNS algorithm. It can be inferred that for code variable attacks, combinatorial optimization is inefficient when the substitutes set of variables is relatively large.
> 2.	We hash adversarial samples corresponding to each variable into 16 buckets through the LSH algorithm. To improve ASR, each bucket is sampled up to 6 times. Assuming that a piece of code has 6 variables, the maximum number of attacks is 576. As shown in the table above, RNNS’s QT is 1.8-2.2 times that of LSH, and the QT index has dropped significantly. However, at the same time, LSH’s ASR index is inferior to RNNS by 42.56%-54.13%. One possible reason for LSH causing low ASR is that the distribution of adversarial samples in each bucket is uneven.
>
> We did not conduct comparative experiments on Blockwise Bayesian Attack [5] for the main reason:
> To achieve a scalable Bayesian optimization algorithm, Blockwise Bayesian Attack decomposes an input sequence into disjoint blocks. However, for code, the same variable may be distributed anywhere in the code, making it difficult to partition.
>
> **For Q2**:
>
> We do not use the six datasets in the experiments for the victim models. The variable name encoder is pre-trained on [CodeSearchNet]( https://huggingface.co/datasets/code_search_net) by masked language modelling independently.
>
> **For Q3**:
>
> 1. There are some differences between adversarial attacks on code variables and those on natural language: code segments are generally longer, and the substitutes set of code variables is much larger than the synonym set of natural language words. The impact of variable replacement on code semantics is often smaller than that of word replacement on natural language semantics. Therefore, some combinatorial optimization algorithms that are highly efficient for natural language attacks face many challenges in code variable attacks.
> 2. LSH is an effective means to significantly reduce QT. However, in the field of code variable attacks, how to achieve a uniform distribution of adversarial samples in each bucket is a topic that needs further research.
>
>  We will supplement more detailed content in the revised version.

---

### Official Review · Reviewer_Wyoa · 2023-08-06

**Soundness:** 3

**Excitement:**

3: Ambivalent: It has merits (e.g., it reports state-of-the-art results, the idea is nice), but there are key weaknesses (e.g., it describes incremental work), and it can significantly benefit from another round of revision. However, I won't object to accepting it if my co-reviewers champion it.

**Paper Topic And Main Contributions:**

This paper proposes a search-based black-box adversarial attack method to create challenging adversarial samples based on the search seed vector in the variable representation space. This method can greatly reduce the verification cost of the substitute, remove the restrictions on the size and diversity of the substitute set, and achieve a significant improvement in terms of ASR without increasing QT. And it also can construct a large-scale, diverse and real substitute set at low cost.The adversarial examples from RNNS can be used to improve the model robustness.

**Questions For The Authors:**

1.	What are ExtractVar and RankVarsWithUncertainty exactly in section3.2.1? What are the specific processes?

2.	In Table2, on some data sets, the query time of this method is much higher. Why is this result not discussed? Is this the reason for the high attack success rate?


**Reasons To Accept:**

1.	The motivation of the article is relatively clear, pointing out three limitations of current research and giving solutions for each limitation.

2.	The strategy the article proposed for selecting the substitute variable from substitute sets has a certain degree of innovation and rationality to achieve lower cost.


**Reasons To Reject:**

1.	The abstract is too long to understand. It does not show the key points and advantages of the article.

2.	Many specific details are missing in the method part, and the specific process of each module is not clearly explained.



**Reproducibility:**

3: Could reproduce the results with some difficulty. The settings of parameters are underspecified or subjectively determined; the training/evaluation data are not widely available.

**Reviewer Confidence:**

3: Pretty sure, but there's a chance I missed something. Although I have a good feel for this area in general, I did not carefully check the paper's details, e.g., the math, experimental design, or novelty.

---

> ### Author Rebuttal · Authors · 2023-08-29
>
> Thank you for recognizing the validity and innovation of our method. We appreciate your valuable feedback. Due to space constraints in the main text, we did not provide a detailed description. We will refine it further in the final version, including the abstract.  We have carefully read your questions and provide the following answers.
>
> **For Q1**:
>
> **ExtractVar** implementation consists of three steps:
> 1. Use the open-source tool tree-sitter to extract all variables in the current dataset and perform deduplication filtering.
> 2. For each legal variable, we tokenize the variable and then take the embedding of each token through the variable-name encoder pre-trained on [CodeSearchNet](https://huggingface.co/datasets/code_search_net) independently by masked language modelling, and perform mean pooling operation on tokens to obtain the embedding of variable.
> 3. Save all variables and their corresponding embeddings to obtain the initial substitutes set.
>
> **RankVarsWithUncertainty** implementation consists of five steps:
> 1. Specify a fixed variable array, named, $VarArray$. In this experiment, the size of this variable array is 16, and the variable length ranges from 1 to 5.
> 2. For a certain variable in the code, we replace it with each variable in $VarArray$ in turn, so that 16 mutated codes are obtained.
> 3. Query the victim model with these 16 mutated codes to obtain the probability distribution of each category.
> 4. Calculate the variance of each category probability distribution.
> 5. Calculate the mean of all category variances as the uncertainty of variable.
> The greater the uncertainty of a variable, the more valuable it is for attack.
>
> **For Q2**:
>
> In Table 2, the three groups of experiments with the most attack attempts are Clone+GraphCodeBert, Java250+CodeT5, and Authorship+CodeBert, with ASRs of 41.28%, 63.80%, and 73.39%, respectively, but they are not the highest attack success rates. On the contrary, the experiment with the highest ASR is Defect+CodeT5, which has an ASR of 89.45%, but its QT is the smallest among all experiments. Therefore, there is no absolute causal relationship between QT and ASR.

---

### Meta-Review · Area_Chair_4U2a · 2023-09-19

**Recommendation:** 3

**Metareview:**

The reviewers agree that the paper provides sufficient support for its major claims/arguments, but were ambivalent about the merits and readiness. The main strengths of the paper are clear motivation, model setup relevant to the application, experimental performance, choice of comparators. The excitement is diminished mainly because of lack of clarity in methodology that left the readers wonder what were the exact contributions of the paper.

---

### Decision · Program_Chairs · 2023-10-07

**Decision:**

Accept-Findings

**Comment:**

The reviewers agree that the paper provides sufficient support for its major claims/arguments, but were ambivalent about the merits and readiness. The main strengths of the paper are clear motivation, model setup relevant to the application, experimental performance, choice of comparators. The excitement is diminished mainly because of lack of clarity in methodology that left the readers wonder what were the exact contributions of the paper.